# Development and Validation of a Rapid Lateral Flow E1/E2-Antigen Test and ELISA in Patients Infected with Emerging Asian Strain of Chikungunya Virus in the Americas

**DOI:** 10.3390/v12090971

**Published:** 2020-09-01

**Authors:** Ankita Reddy, Irene Bosch, Nol Salcedo, Bobby Brooke Herrera, Helena de Puig, Carlos F. Narváez, Diana María Caicedo-Borrero, Ivette Lorenzana, Leda Parham, Kimberly García, Marcela Mercado, Angélica María Rico Turca, Luis A. Villar-Centeno, Margarita Gélvez-Ramírez, Natalia Andrea Gómez Ríos, Megan Hiley, Dawlyn García, Michael S. Diamond, Lee Gehrke

**Affiliations:** 1E25Bio, Cambridge, MA 02139, USA; areddy@e25bio.com (A.R.); nsalcedo@e25bio.com (N.S.); bbherrera@e25bio.com (B.B.H.); lgehrke@mit.edu (L.G.); 2Institute for Medical Engineering & Science, Massachusetts Institute of Technology, Cambridge, MA 02139, USA; hpuig@mit.edu (H.d.P.); meganjhiley@gmail.com (M.H.); gdawlyn@gmail.com (D.G.); 3Perelman School of Medicine, University of Pennsylvania, Philadelphia, PA 19104, USA; 4Department of Medicine, Mount Sinai School of Medicine, New York, NY 10029, USA; 5Department of Immunology and Infectious Diseases, Harvard T.H. Chan School of Public Health, Boston, MA 02115, USA; 6Wyss Institute for Biologically Inspired Engineering, Harvard Medical School, Boston, MA 02115, USA; 7Programa de Medicina, Facultad de Salud, Universidad Surcolombiana, Neiva, Huila, Colombia; cfnarvaez@usco.edu.co; 8Departamento de Salud Pública y Epidemiología de la Pontificia Universidad, Javeriana Cali y Escuela de Salud Pública de la Universidad del Valle, Cali, Colombia; diam121@hotmail.com; 9Instituto de Investigación en Microbiología, Universidad Nacional Autónoma de Honduras, Tegucigalpa, Honduras; ivettelorenzana@yahoo.com (I.L.); lparham29@hotmail.com (L.P.); kimfa_2010@hotmail.com (K.G.); 10Dirección de Investigación en Salud Pública, Instituto Nacional de Salud, Bogotá, Colombia; mmercado@ins.gov.co; 11Laboratorio de Virología, Dirección de Redes en Salud Pública, Instituto Nacional de Salud, Bogotá, Colombia; arico@ins.gov.co; 12Departments of Escuela de Medicina, Universidad Industrial de Santander and AEDES Network, Bucaramanga, Santander, Colombia; luisangelvillarc@gmail.com (L.A.V.-C.); margarita.gelvez@hotmail.com (M.G.-R.); natagomezr@gmail.com (N.A.G.R.); 13Departments of Medicine, Molecular Microbiology, Pathology & Immunology, Washington University School of Medicine, Saint Louis, MO 63110, USA; mdiamond@wustl.edu; 14Department of Microbiology and Immunobiology, Harvard Medical School, Boston, MA 02115, USA

**Keywords:** Chikungunya fever, ELISA, lateral flow, E1/E2 antigen detection, alphavirus, Latin America, acute phase diagnosis, rapid diagnosis

## Abstract

Since its 2013 emergence in the Americas, Chikungunya virus (CHIKV) has posed a serious threat to public health. Early and accurate diagnosis of the disease, though currently lacking in clinics, is integral to enable timely care and epidemiological response. We developed a dual detection system: a CHIKV antigen E1/E2-based enzyme-linked immunosorbent assay (ELISA) and a lateral flow test using high-affinity anti-CHIKV antibodies. The ELISA was validated with 100 PCR-tested acute Chikungunya fever samples from Honduras. The assay had an overall sensitivity and specificity of 51% and 96.67%, respectively, with accuracy reaching 95.45% sensitivity and 92.03% specificity at a cycle threshold (Ct) cutoff of 22. As the Ct value decreased from 35 to 22, the ELISA sensitivity increased. We then developed and validated two lateral flow tests using independent antibody pairs. The sensitivity and specificity reached 100% for both lateral flow tests using 39 samples from Colombia and Honduras at Ct cutoffs of 20 and 27, respectively. For both lateral flow tests, sensitivity decreased as the Ct increased after 27. Because CHIKV E1/E2 are exposed in the virion surfaces in serum during the acute infection phase, these sensitive and specific assays demonstrate opportunities for early detection of this emerging human pathogen.

## 1. Introduction

Chikungunya virus (CHIKV) is an increasingly prevalent alphavirus that is transmitted by Aedes mosquitoes [1]. In recent years, CHIKV has re-emerged at an unprecedented rate, spreading to over 100 countries across five continents and producing over one million infections annually [2,3,4]. As mosquito breeding grounds expand in response to climate change and globalization, CHIKV infections are expected to pose an even greater threat to public health.

Chikungunya fever, which is caused by infection of CHIKV, is a debilitating disease which often includes joint pain and high fever, and a plethora of additional nonspecific symptoms including rash, abdominal pain, vomiting, diarrhea, myalgia, and headache. Although the acute clinical manifestations often subside after 1−3 weeks, chronic joint pain lasting months to years can significantly impair movement and undermine quality of life [5]. Severe forms of Chikungunya fever also include neurological complications, myocarditis, pneumonia, lymphadenopathy, hepatitis, and pancreatitis [2]. Treatment for Chikungunya fever is mainly supportive and symptomatic, but early diagnosis is vital to enabling care and preventing further complications that can be debilitating and life-threatening. Early diagnosis also allows patient triaging and infection surveillance for timely care and disease prevention, particularly during outbreaks.

CHIKV is difficult to diagnose solely through clinical findings due to the nonspecific nature of the febrile diseases symptoms [6,7]. The nonspecific symptoms overlap with dengue (DENV) and Zika (ZIKV) viruses—diseases that often co-circulate with CHIKV—rendering accurate diagnosis particularly complex during the first days of disease [6,8,9,10]. Because disease outcomes and supportive treatment significantly differ between these three diseases, accurate diagnosis is critical to outbreak control, surveillance, and prevention [11]. Accurate diagnosis is also significant for research related to vaccine efficacy and drug development.

CHIKV contains an 11.8 kb positive-sense, single-stranded RNA genome. The virus encodes four conserved nonstructural proteins (nsP 1–4), a capsid protein, two envelope glycoproteins (E1 and E2), and two cleavage products (E3 and 6K) [12]. The E1 and E2 proteins offer an ideal target for diagnosis because they are secreted at high concentrations into human blood during the acute phase of infection when viremia is high.

Presently, there is an urgent need for an accurate and early diagnosis during the acute phase for CHIKV-infected patients to enable rapid clinical response and appropriate epidemiological surveillance. Currently available methods of diagnosis include viral isolation, polymerase chain reaction (PCR) [13,14,15,16,17,18], reverse transcription loop-mediated isothermal amplification (RT-LAMP), and serological tests such as IgM/IgG lateral flows, enzyme-linked immunosorbent assay (ELISAs), and indirect immunofluorescent assays (IIFAs) [19,20,21,22]. All of these assays contain significant barriers to enabling appropriate outbreak response, ranging from high costs and lengthy testing times to post-acute phase diagnosis.

In this study we describe the development and performance of two methods of diagnosis that enable early and accessible diagnosis: an E1/E2 antigen-based test in both an ELISA and rapid lateral flow format. Our data indicates high specificity and sensitivity of the tests using infected samples from the CHIKV endemic regions of Honduras and Colombia, areas severely underrepresented in previous studies.

## 2. Materials and Methods

### 2.1. Study Design

This study aimed to develop a CHIKV antigen-based ELISA and lateral flow tests and to validate the accuracy of the assays using PCR-confirmed acute fever serum samples. All tests were performed on-site at the University of Tegucigalpa in Honduras and the Instituto Nacional de Salud in Colombia. All limit of detection experiments were conducted in a biosafety level (BSL) 3 laboratory at the Ragon Institute.

### 2.2. Clinical Samples

A total of 129 acute Chikungunya fever clinical serum samples and 60 negative patient samples were used in this study. Of these samples, 100 fever samples and 60 negative samples were collected by medical personnel from Honduras at the University of Tegucigalpa in Honduras. The remaining 29 acute fever samples were collected by medical personnel at the Instituto Nacional de Salud in Colombia. All clinical serum samples were de-identified and collected during the acute phase (1 to 5 days after the onset of illness). All patients in the Honduras cohort were tested for dengue and Zika virus infections by PCR. The single infections from Chikungunya positive patients were selected. Further confirmation of Chikungunya virus resulted from negative dengue and Zika antigen tests available in the laboratory. All patients from each of the cohorts provided informed consent for the original collection of the samples. The primary studies under which the samples and data were collected received an exemption determination from the Massachusetts Institute of Technology Internal Review Board (IRB) and local research ethics committees at University of Tegucigalpa, Honduras (Comite Etico en Investigación Biomedica, ID: NMRCD.2010.0010) and Instituto Nacional de Salud in Colombia (INS Ethics Committee, ID: CTIN-31–2015).

### 2.3. Antibody Production and Selection 

CHIKV antibodies for Combination A (48 and 155) were generated previously in mice [23]. CHIKV antibodies for Combination B were produced by immunizing a C57BL/6 mouse with CHIKV virus-like particles (Native Antigen CHIKV-VLP Q5XXP3.1). A total of 1056 antibodies were harvested from the hybridomas. To maximize sensitivity and sensitivity, antibodies were produced and selected through mouse immunization and a set of two screening methods. The binding of these antibodies to CHIKV VLP was measured by ELISA. Response was measured as fold above background, by subtracting the negative control OD_450_ from the OD_450_ of interest, and then dividing by the negative control OD_450_. The binding of antibodies to the genetically related Mayaro virus (MAYV) VLP (Native Antigen MAYV_VLP AJA37502.1) was measured for counter-screening. The 48 antibodies with the highest fold above background to CHIKV VLP but with low binding affinity to MAYV VLP using the primary ELISA screen underwent a secondary screen by fluorescent-activated cell sorting (FACS), designed to evaluate the recognition of monoclonal antibodies to CHIKV-infected Vero cells. The strain utilized to infect Vero cells was the East Central South African (ECSA) genotype (KX228391). The antibodies that stained positively by flow cytometry on infected cells were isotyped and purified using Protein L or Protein G according to their light chain binding epitopes. The antibody pairs were evaluated on a dipstick format and selected based upon the lowest limit of detection and dissociation constant through image analysis, as adapted from Bosch et al. [24].

### 2.4. ELISA for the Detection and Quantification of CHIKV E1/E2

To validate the E1/E2 ELISA, 100 fever samples and 60 negative samples from Honduras were tested using an adapted protocol from Bosch et al. [24]. Detection antibodies were first biotinylated using ThermoFisher Scientific EZ-Link Sulfo-NHS-LC-Biotinylation Kit, according to the manufacturer’s instructions (cat no. 21335, Pierce Biotechnology, Waltham, MA, USA). To prepare the ELISA, ninety-six-well CoStar flat bottom high binding plates (cat. no. 3590, Corning, Corning, NY, USA) were coated with 100 µL of the specific antibody (mAb 155) at a 1 µg/mL, diluted in 1X PBS pH 7.4 (cat no. 10010031, Gibco, Gaithersburg, MD, USA). After incubating the plates overnight at room temperature, the antibody was discarded and each well was incubated for 2 h at room temperature with 200 µL/well of 5% Blotto (cat. no. sc−2325, Santa Cruz Biotechnology, Santa Cruz, CA, USA) made from 5% nonfat dry milk (cat. no. sc−2325, Santa Cruz Biotechnology) and 0.05% Tween 20 (cat. no. *p*−1379, Sigma-Aldrich, St. Louis, MO, USA) diluted in PBS. After discarding the liquid, 50 µL of serum sample diluted in 50 µL 2.5% Blotto in PBS were incubated in each well for 1 h at room temperature. After washing the plates three times with 0.1% Tween 20 in PBS, 100 µL/well of biotin-labeled mAb 48 at 1 µg/mL was incubated for 1 h at room temperature. The plates were washed four times with the 0.1% Tween 20 solution. One hundred µL/well of peroxidase-labeled streptavidin High Sensitivity (cat. no. 21130, Thermo-Fisher Scientific, Waltham, MA, USA) at 1:1000 dilution, diluted in 2.5% Botto, was added and incubated for 1 h at room temperature. The plates were again washed four times with the 0.1% Tween 20 in PBS. Following the wash steps, 100 µL/well of tetramethylbenzidine single solution (cat. no. 002023, Life Technologies, Carlsbad, CA, USA) were pipetted into each well to develop the color reaction and stopped by the addition of 50 µL/well of 2M sulfuric acid (cat. no. 8315–32, Ricca Chemical Company, Arlington, TX, USA). The plates were read by a TriStar LB 941 spectrophotometer (Berthold Technologies) at a wavelength of 450 nm.

### 2.5. Lateral Immune Detection Methods for the Quantification of CHIKV E1/E2

The E1/E2 lateral flow test was validated using 29 fever serum samples, of which 19 samples were from Honduras and 10 were from Colombia. Dipstick and lateral flow assays were constructed in the lab using an adapted protocol. Briefly, forty-nanometer gold nanoparticles (Innova Biosciences, Cambridge, UK) were conjugated to the CHIKV antibodies according to the manufacturer’s instructions. The antibody was first diluted to 0.1 mg/mL in the supplied dilution buffer. Next, 12 µL of diluted antibody were mixed with 42 µL of reaction buffer. Forty-five microliters of the mix were then used to suspend the lyophilized gold nanoparticles (OD_20_). The antibody-nanoparticle mix was incubated for 10 min at room temperature, followed by the addition of 5 mL of quencher solution to stop the coupling reaction. After adding the quencher solution, 100 mL of 1% Tween 20 in PBS and 50 mL of 50% sucrose in water were added to the conjugates before use in immunochromatography. Dipsticks were used to screen antibodies and collect limit of detection values. Lateral flows were used to collect limit of detection values and test patient samples in the field.

### 2.6. Image Analysis

Lateral flow tests were analyzed through image analysis to quantify the signal intensity on the strip. Following test runs, the strips were machine scanned and converted to greyscale. ImageJ software was used to quantify the signal in the test area and in the positive control area. The normalized signal was computed as a ratio of the test to positive control, and this data was used in LoD calculations and ROC analyses. 

### 2.7. Limit of Detection (LoD) Analysis

Functional antibody pairs were defined through combinatorial dipstick trials. One antibody was conjugated to gold nanoparticles and one antibody was adsorbed to nitrocellulose membrane. The resulting nanoparticle conjugates-membrane pairs were tested using the CHIKV virus-like particles (VLP) and MAYV VLP as a counter-screen. These proteins were present at a concentration of 150 ng/mL in the testing. The signal on the dipstick is measured on a scale from 0 (no visible signal) to 1 (strongest signal) to measure antibody sandwich binding to VLP. Antibody pairs were screened against Colombian and Honduras CHIKV strains. Envelope (E1) sequences were obtained and confirmed to belong to the Asian genotype.

Two antibody combinations were selected from this testing matrix to perform further limit of detection experiments in an ELISA format. R software was used to calculate the limits of detection (LoD) and dissociation constant (K_d_) for each antibody pair in each diagnostic format. The K_d_ was derived by keeping each antibody concentration constant and running decreasing concentrations of either CHIKV E1 and E2 protein or CHIKV virus-like particles (VLP). The results were fitted using a Langmuir equation, gray_n_ = [antigen]/K_d_ + [antigen], where gray_n_ is the normalized signal intensity on the lateral flow, [antigen] is the concentration of E1/E2 or VLP and K_d_ is the effective binding constant in a Langmuir-like system. The LoD was measured as the concentration of E1/E2 or VLP that displayed a signal 5 times the value of the standard deviation of the negative control. 

### 2.8. RNA Extraction and Quantitative RT-PCR

Samples were collected and processed for RNA extraction followed by RT-PCR in each of the participating laboratories. RNA was extracted according to the QIAamp Viral RNA Mini Handbook for purification of viral RNA from plasma, serum, cell-free body fluids and culture supernatants (cat no. 52904, Qiagen, Hilden). Virus identity including serotypes was determined using quantitative PCR. The Fast Track Diagnostics Dengue/Chik real-time PCR protocol and reagents were used to process samples in India, according to the manufacturer’s instructions (TaqMan). The Agpath-ID One-Step Real-Time PCR protocol and reagents were used to process samples in Honduras and the CDC Trioplex Real-Time PCR protocol and reagents were used to process samples in Colombia, according to the manufacturers’ instructions. 

### 2.9. Receiver Operator Characteristic (ROC) Analysis

GraphPad Prism 8.0 software was used to report the performance of the ELISA, receiver operating characteristic (ROC) curves. The ROC curve presents test performance as true positive rate (% sensitivity) versus false positive rate (100%—% specificity). Optimal cutoff values, which maximize sensitivity and specificity, were calculated from the ROC curve also using GraphPad Prism 8.0. The sensitivity is defined as the fraction of total confirmed positive samples that are true positives according to the test. The specificity is defined as the fraction of total confirmed negative samples that are true negatives according to the test. Confidence intervals (CI) using the Wilson/Brown method and area under curve (AUC) were calculated for each serotype and PAN using GraphPad.

## 3. Results

### 3.1. Antibody Selection for CHIKV ELISA and Lateral Flow Assays

Of 1056 antibodies harvested from the CHIKV-immunized mice, mAb 48 with mAb 155 (Combination A) and mAb 4 with mAb 340 (Combination B) were selected for the sandwich ELISA and lateral flow tests. The ELISA screening of the 48 chosen antibodies displayed low or nonexistent detection for MAYV VLP across all antibody clones, with 73% of the clones presenting at least 20 fold above background (Appendix A). We show that of the ELISA positive clones, 16 out of 41 clones tested positive using FACS, corresponding to 39% positivity (Appendix A). The two antibody pairs, Combination A and B, were chosen based upon the high CHIKV VLP binding affinity and discrimination between CHIKV VLP and MAYV VLP determined from the ELISA and flow cytometry of infected cell screening as well as the low limits of detection. 

### 3.2. Limits of Detection 

Combinatorial dipstick analysis was first performed to select the antibody combinations, demonstrating high binding affinity in a rapid test format (Appendix A). From this analysis we chose mAb 48 with mAb 155 and 4 with 340 to measure the limit of detection on an ELISA format. The limits of detection and dissociation constants of both antibody pairs (48 with 155; 4 with 340) for the detection of CHIKV E1 and E2 were calculated by ELISA, dipstick, and/or lateral flow. The limits of detection were between 37.08 and 844.16 ng/mL (Appendix A, Appendix A) which is within the viral load concentration found in acute Chikungunya patients [7,17].

### 3.3. Performance of CHIKV E1/E2 ELISA 

The E1/E2 CHIKV ELISA was validated using 100 PCR-confirmed Chikungunya samples and 60 negative samples from Honduras. Sensitivity is defined as the fraction of true positive test results from the population of PCR-positive samples. Specificity is defined as the fraction of true negative test results from the samples that were PCR-negative for the tested serotype. The sensitivity and specificity of the developed E1/E2 ELISA were determined using various cycle threshold (Ct) value cutoffs (Table 1). Ct value cutoffs represent the number of PCR cycles at which generated fluorescence crosses a threshold during the linear phase of amplification. The OD_450_ was inversely related to the Ct value, with a linear regression line of y = −0.04908x + 1.731 and *p* < 0.0001 (Figure 1A). The overall sensitivity and specificity was 51% and 96.67%, respectively. Across the Ct range of 20 to 35, the performance of the ELISA ranged from 41.33% to 95.45% in sensitivity and 84.76% to 98.02% in specificity, with an area under the ROC curve (AUC) range of 0.61 to 0.94 (Table 1, Figure 2). A Ct value cutoff of 22 maximized the performance of the ELISA, which has a sensitivity of 95.45% and specificity of 92.03%. The optimal ELISA OD_450_ cutoff value, in which the sum of the sensitivity and specificity was maximized, was 0.22 across all Ct values. Based on the performance analysis across Ct values, the CHIKV E1/E2 ELISA demonstrated high sensitivity and specificity, particularly for low Ct cutoff values.

### 3.4. Performance of CHIKV E1/E2 Lateral Flow

Our lateral flow assay was used to detect CHIKV E1/E2 using either the mAbs 48 and 155 combination (Combination A), or mAbs 4 and 340 (Combination B). In total, 29 CHIKV samples were used from Honduras and Colombia to validate the lateral flow tests, of which the 19 samples from Honduras were used in both versions. The intensity of signal on the lateral flow was found to be inversely correlated with the Ct value (Figure 1B–D). The slopes of the linear regression line for Combination A in Honduras, Combination B in Honduras, and Combination B in Colombia were 0.07238, −0.05481, and −0.034, respectively. The *p* values were also <0.0001, <0.0002, <0.0182. When comparing the two antibody combinations in the lateral flow format side-by-side using the same Honduras samples, both versions performed optimally at a Ct cutoff of 27 with 100% sensitivity and 100% specificity (Figure 3A,B and Table 2A,B). At this Ct value, the optimal ELISA OD_450_ cutoff was 0.57 and 0.53 for Combination A and Combination B, respectively. Across the Ct range of 20 to 29, the performance of Combination A ranged from 55.56% to 100% in sensitivity and 90.00% to 100% in specificity, with an AUC range of 0.78 to 1. The sensitivity and specificity of Combination B ranged from 55.56% to 100% and 83.33% to 100%, respectively, with an AUC range of 0.78 to 1 as well. 

A total of ten samples from Colombia were used to validate the lateral flow test made from Combination B antibodies (Figure 3C, Table 2C). Data using Combination A antibodies was not collected. From this cohort, the sensitivity and specificity ranges were 85.71% to100% and 77.78% to 100%, respectively. The optimal lateral flow intensity cutoff ranged from 0.614 to 0.6289 and the AUC range was 0.78 to 1. The test performed optimally at a Ct cutoff of 20 with 100% sensitivity and specificity. Across Honduras and Colombia, the Combination B test performed comparably with Combination A, demonstrating a high sensitivity and specificity over a range of Ct cutoff values.

## 4. Discussion

In this study we present two assays—an antigen-based ELISA and lateral flow test—which are alternatives to relying on RT-PCR methods or serology for the timely detection or diagnosis of CHIKV infections. The rate of CHIKV infections has markedly increased within the past two decades, yet clinical diagnostic methods remain impractical for public health response. Currently, there are no commercially available methods for early high-throughput screening or rapid tests for CHIKV. Diagnosis by PCR is common and can be used to diagnose early stages of fever, yet remains costly in terms of time, labor, and resources. PCR also requires experienced personnel for sample preparation and nucleic acid extraction. This method is also prone to error due to the degradation of RNA and the potential for amplicon contamination between samples. Serological methods, such as an IgG/IgM-based ELISAs, are also common yet are limited to post-acute phase diagnosis [25,26,27,28,29]. Studies show that these tests perform extremely poorly during the first week of fever, as IgM levels are detectable only between day 4 and 7 after illness onset [1]. IgM assays also remain impractical for early detection as these antibodies can exist in the body months after illness [28].

There currently exist few reports of antigen-based assays for CHIKV. Shukla et al. developed an antigen-capture ELISA by producing hyperimmune sera from mice and rabbits and using the antibodies in a sandwich ELISA format [28]. Although the test was in 96% concordance with RT-PCR, the samples were collected from CHIKV patients in India, thus limiting the performance of the test to the detection of the CHIKV Asian lineage [30]. An immunochromatographic assay was reported by Okabayashi et al. (2015) in which monoclonal antibodies against the E1 protein were developed and used to detect the CHIKV antigen [31,32,33]. However, this test targets only one envelope protein and displays a limit of detection of 1 × 105 PFU/mL. RT-LAMP methods have also been considered as an acute-phase diagnostic tool, particularly due to their cost-effectiveness compared to PCR. However, electricity, trained personnel, and time (at least one hour) are required, limiting the capacity for large-scale screening as well as diagnosis in low-resource areas [34]. 

Our study introduces an antigen ELISA and lateral flow tests with high sensitivity and specificity for the detection of both CHIKV E1 and E2—addressing the dearth of antigen-based tests and the several limitations present in previously reported assays. We performed an extensive antibody screening to maximize the performance of antigen capture, as well as high sensitivity and specificity in two different assay formats across a range of PCR Ct cutoff values, highlighting the potential for the broad application of these assays. Moreover, our assays are tested with samples from Latin America, which has previously been underrepresented in the validation of the previous antigen-based tests. Latin American countries, particularly Honduras and Colombia, bear a sizable burden of dengue, Zika, and Chikungunya infections—three mosquito-borne diseases which share several nonspecific symptoms, often leading to misdiagnosis and uncertainty. After the introduction of Chikungunya to the Caribbean in 2013, the disease has rapidly spread in the Americas, with 998,015 cases reported in 2016 alone [35,36]. Our CHIKV antigen-based assays development and validation are significant because they display high sensitivity and specificity in samples from Latin America. The lateral flow tests reached maximum sensitivity and specificity over a Ct value range of 20−27 and 21−27 for Colombia and Honduras, respectively. Our lateral flow tests reached 100% sensitivity and specificity using either of the two antibody pairs selected from screening. The ELISA format reached sensitivity and specificity reaching 95.45% and 98.02%, respectively, with a Ct cutoff of 22. However, the accuracy significantly decreases as the Ct values rise for all three assays.

The main significance of these antigen-based CHIKV diagnostic assays is the ability to detect the virus within the first 5 days of fever onset. Although overall sensitivity is low, our analysis shows that that accuracy significantly increases when individuals have a higher viral load—which correlates to peak infectivity and progression to severe symptoms [37,38,39]. Sensitivity was likely affected by samples with low viremia, possibly because sample collection took place in the latter end of the optimal diagnostic window or some individuals had lower viremia than others. Samples with a Ct value close to 35, for instance, are more difficult to detect than samples with a low Ct value due to the lower viremia. As shown by the linear regression models of assay signals in relation to the Ct values, the signal decreases with increasing Ct values. Thus, the assays perform more accurately with samples with more virus and lower Ct values. Moreover, in Honduras, temporary temperature fluctuations of stored samples possibly led to protein degradation, affecting sensitivity values. Nevertheless, the AUC remained close to 1 for all countries including Honduras, for both the lateral flows and ELISA.

There is opportunity to expand upon this work as well. Future validation tests will aim to include patient samples from Asia and Africa to expand validation of the tests with different CHIKV lineages. In developing these assays, we chose to select for antibodies that discriminated between CHIKV and MAYV particularly because Mayaro virus is also endemic to areas in Latin America. Future studies can expand upon this screen to include more related alphaviruses such as O’nyong’nyong virus, found in Sub-Saharan Africa, and Ross River virus, found in Australia and the Pacific island regions. Future studies should also aim to analyze assay performance with inactivated serum samples such that point-of-care testing can be used with noninfectious material particularly in areas with limited laboratory capacity.

The dual opportunities presented by highly specific and sensitive antigen-based lateral flow tests and ELISAs for CHIKV diagnosis are significant. The lateral flow test offers an extremely low-cost and rapid method for early detection in both low and abundant resource settings. The ELISA offers a method for high-throughput screening that is more accessible than PCR. Particularly during outbreaks, both assays may enable systems of hospital triage and disease surveillance that better equip public health response. When the disease is detected within the first five days of fever, early, supportive treatment may be administered to prevent progression to severe forms of CHIKV such as neurological disorders, durable joint pain, and death. Moreover, an early diagnosis enables more targeted care, avoiding the large costs that hospitals incur from a late diagnosis or misdiagnosis. Early detection of infection can also provide critical data to prevent further transmission and alarm surveillance systems.

Ideally, these antigen-based tests may be used in conjunction with serology-based tests, particularly when days of patient fever are uncertain or lie between diagnostic windows. Taken together, this study describes the development and validation of a highly specific and sensitive CHIKV E1/E2 rapid lateral flow and ELISA assay. These antigen-based assays are a crucial component to enabling early detection of disease for proper and timely care, economic allocation of clinical resources, and adequate epidemiological measures.

## Figures and Tables

**Figure 1 viruses-12-00971-f001:**
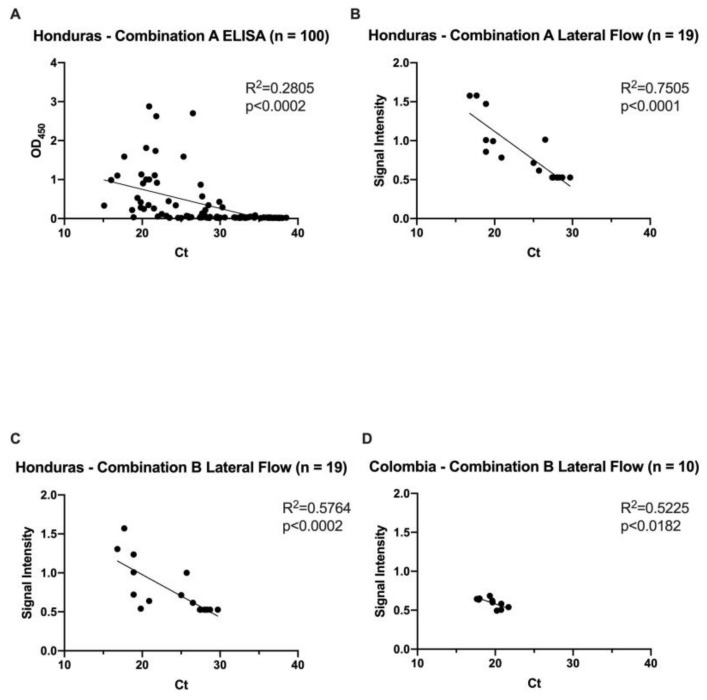
Chikungunya virus (CHIKV) E1/E2-based enzyme-linked immunosorbent assay (E1/E2 ELISA) and Lateral Flow Test compared to qPCR. The OD_450_ or signal intensity of samples tested through ELISA (**A**) and lateral flow (**B**–**D**), respectively, are shown in relation to the PCR Ct values. The tests used either antibody Combination A (48 and 155) or Combination B (4 and 340). The *p* values are <0.0001 (**A**), <0.0001 (**B**), <0.0002 (**C**), <0.0182 (**D**).

**Figure 2 viruses-12-00971-f002:**
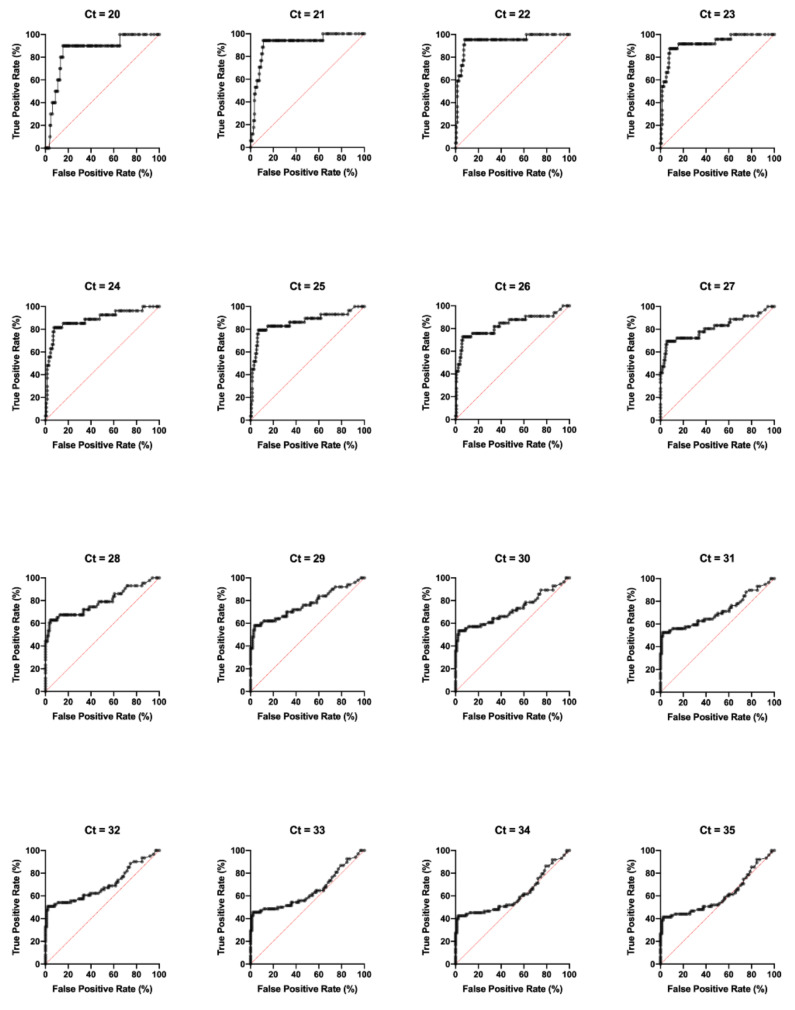
Performance of CHIKV E1/E2 ELISA Test. Receiver operating characteristic (ROC) curve analysis of PCR-tested patient samples from Honduras. Incremental Ct values are used to characterize positive and negative samples, to which ELISA performance is compared. Test performance is demonstrated in terms of true positive rate (%) versus false positive rate (%).

**Figure 3 viruses-12-00971-f003:**
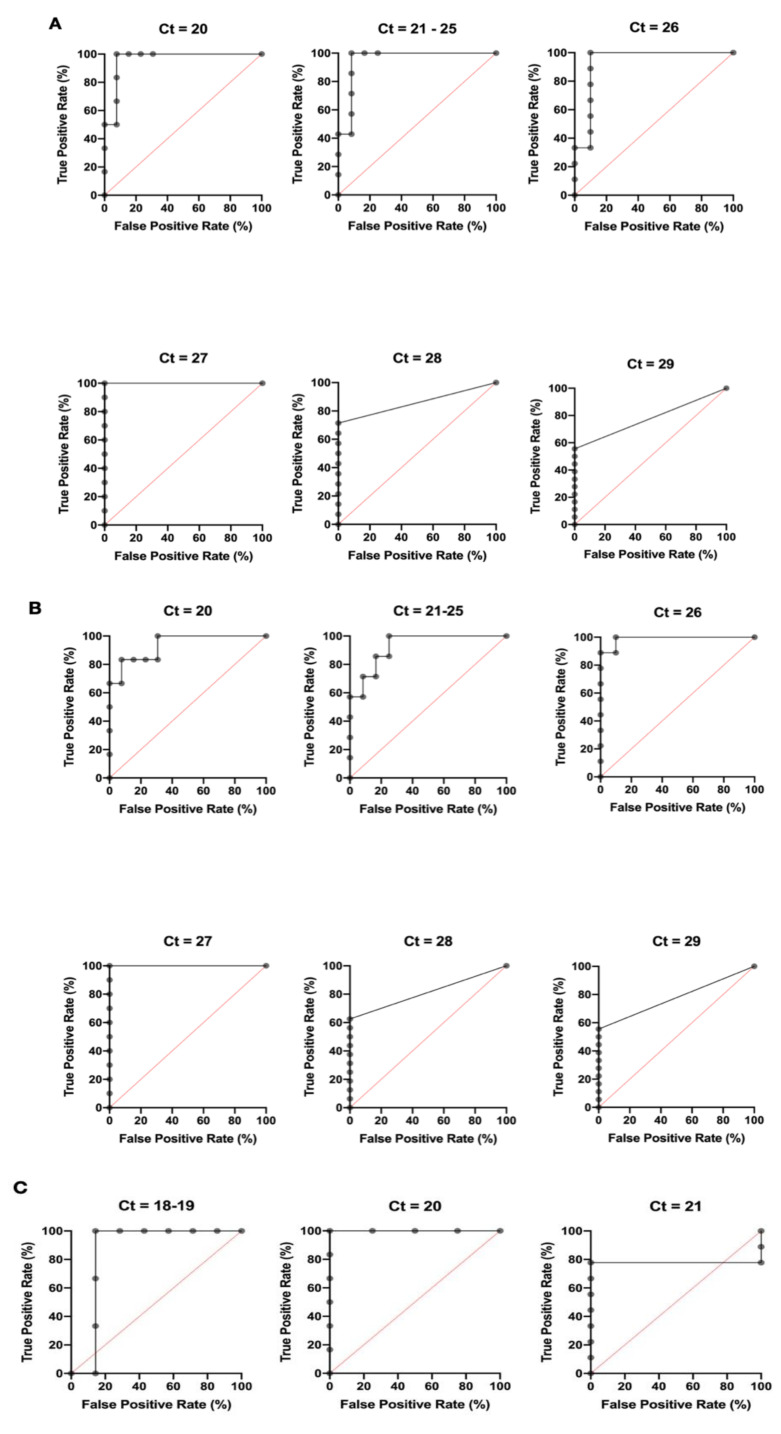
Performance of CHIKV Lateral Flow Test. Receiver operating characteristic (ROC) curve analysis of PCR-tested patient samples. The lateral flow test was constructed with Combination A antibodies using 19 samples from Honduras (**A**), Combination B antibodies using 19 samples from Honduras (**B**), and Combination B antibodies using 10 samples from Colombia (**C**). Incremental Ct values are used to characterize positive and negative samples, to which lateral flow test performance is compared. Test performance is demonstrated in terms of true positive rate (%) versus false positive rate (%).

**Table 1 viruses-12-00971-t001:** Performance of CHIKV E1/E2 ELISA Test. Area under curve (AUC) of ROC curve, 95% confidence interval (95% CI), optimal OD_450_ cutoff, sensitivity (%), specificity (%), and sample counts were calculated for incremental PCR Ct cutoffs for the E1/E2 ELISA test using antibody Combination A (48, 155).

CHIKV Combination A ELISA Receiver Operator Characteristic (ROC) Analysis (*n* = 160)																
Ct Cutoff	20	21	22	23	24	25	26	27	28	29	30	31	32	33	34	35
AUC	0.86	0.91	0.94	0.92	0.89	0.86	0.83	0.81	0.79	0.77	0.72	0.71	0.70	0.65	0.61	0.61
95% CI	0.74–0.97	0.84–0.99	0.89–1.00	0.86–0.98	0.80–0.97	0.77–0.95	0.74–0.93	0.72–0.91	0.70–0.88	0.68–0.86	0.63–0.82	0.62–0.81	0.61–0.79	0.55–0.74	0.52–0.71	0.52–0.70
OD_450_ Cutoff	0.22	0.22	0.22	0.22	0.22	0.22	0.22	0.22	0.22	0.22	0.22	0.22	0.22	0.22	0.22	0.22
Sensitivity (%)	90.00	94.12	95.45	87.5	81.48	79.31	72.73	69.44	62.79	58.00	53.57	52.54	50.82	45.59	42.47	41.33
Specificity (%)	84.67	88.81	92.03	87.5	92.48	93.13	93.7	94.35	95.73	96.36	97.12	98.02	97.98	97.83	97.7	97.65
N Total Positive	150	143	138	136	133	131	127	124	117	110	104	101	99	92	87	85
N Total Negative	10	17	22	24	27	29	33	36	43	50	56	59	61	68	73	75

**Table 2 viruses-12-00971-t002:** Performance of CHIKV E1/E2 Lateral Flow Test. Area under curve (AUC) of ROC curve, 95% confidence interval (95% CI), signal intensity cutoff, sensitivity (%), specificity (%), and sample counts were calculated for incremental PCR Ct cutoffs for the E1/E2 lateral flow test using antibody Combination A (48 and 155) in Honduras (**A**) and antibody Combination B (4 and 340) in Honduras (**B**) and Colombia (**C**).

A.					
CHIKV Combination A Lateral Flow Receiver Operator Characteristic (ROC) Analysis-Honduras (*n* = 19)					
Ct Cutoff	20	21–24	25–26	27	28
AUC	0.96	0.95	0.94	1.00	0.81
95% CI	0.88–1.00	0.85–1.00	0.83–1.00	1.00–1.00	0.61–1.000
Lateral Flow Signal Intensity Cutoff	0.82	0.75	0.67	0.57	0.57
Sensitivity (%)	100	100	100	100	62.50
Specificity (%)	92.31	91.67	90.91	100	100
N Total Positive	13	12	11	9	3
N Total Negative	6	7	8	10	16
B.					
CHIKV Combination B Lateral Flow Receiver Operator Characteristic (ROC) Analysis-Honduras (*n* = 19)					
Ct Cutoff	20	21–26	27	28	
AUC	0.94	0.93	1.00	0.81	
95% CI	0.82–1.00	0.82–1.00	1.00–1.00	0.61–1.00	
Lateral Flow Signal Intensity Cutoff	0.72	0.53	0.53	0.53	
Sensitivity (%)	83.33	100	100	62.50	
Specificity (%)	92.31	75.00	100	100	
N Total Positive	13	12	9	3	
N Total Negative	6	7	10	16	
C.					
CHIKV Combination B Lateral Flow Receiver Operator Characteristic (ROC) Analysis-Colombia (*n* = 10)					
Ct Cutoff	18–19		20	21	
AUC	0.86		1.00	0.78	
95% CI	0.60–1.00		1.00–1.00	0.51–1.00	
Lateral Flow Signal Intensity Cutoff	0.63		0.59	0.56	
Sensitivity (%)	100		100	77.78	
Specificity (%)	85.71		100	100	
N Total Positive	7		4	1	
N Total Negative	3		6	9	

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
