# Peer review of "Development and Validation of a Rapid Lateral Flow E1/E2-Antigen Test and ELISA in Patients Infected with Emerging Asian Strain of Chikungunya Virus in the Americas"

_viruses, 2020, doi:10.3390/v12090971_

Round 1
Reviewer 1 Report
Manuscript describes two novel assays designed to help diagnose acute infection with Chikungunya virus (CHIKV). This includes an ELISA and lateral flow platforms that involve combinations of isolated monoclonal antibodies reactive to CHIKV E1 and E2 proteins. The authors utilize patient samples from which CHIKV infection is quantitated by qPCR to obtain Ct values. The assays perform very well and are potentially scalable. As such they could have a large impact on rapid and high volume CHIKV diagnosis. Moreover, the same technology could potentially be employed for other alphaviruses in the event that outbreaks such as has occurred with CHKV happen in the future. Two minor comments:
- Lines 208-209, would be more accurate and informative to say that the Ct value cutoffs represent thresholds crossed while in the linear phase of amplification.
- Figure 1 would benefit from presenting the p values and correlation coefficients on the individual charts (e.g. in upper right quadrants).
Author Response
August 23, 2020
Dear Reviewer,
We thank you for your insightful comments to our manuscript entitled Development and validation of a rapid lateral flow E1/E2-antigen test and ELISA in patients infected with emerging Asian strain of Chikungunya virus in the Americas. Please find below our responses to the reviewer comments which detail how we have incorporated the feedback:
Reviewer 1:
- Lines 208-209, would be more accurate and informative to say that the Ct value cutoffs represent thresholds crossed while in the linear phase of amplification.
We have accepted this suggestion and have amended the sentence to “Ct value cutoffs represent the number of PCR cycles at which generated fluorescence crosses a threshold during the linear phase of amplification” (Lines 228-229)
- Figure 1 would benefit from presenting the p values and correlation coefficients on the individual charts (e.g. in upper right quadrants)
We have accepted this suggestion and have added the p value and correlation coefficient to Figure 1.
Thank you and please let me know if there are further comments and/or questions.
Sincerely,
Irene Bosch, PhD
CTO
E25Bio, Inc
ibosch@e25bio.com

Reviewer 2 Report
In this manuscript the authors present a new diagnostic testing methods for Chikungunya virus that would enable early rapid detection of infection.
Line 91 Clinical Samples. More information about the clinical samples needs to be included. Were the patients positive for other potential related viral infections in addition to CHIKV, this needs to be included to rule out the potential for false positives
In the methods, lines 111-114, a second screening method for the antibodies is described using binding of the antibodies to infected cells and analysis by flow cytometry, data needs to be included for this in the results section of the manuscript.
3.1 Additional experiments or information to address the specificity of the diagnostic antibodies needs to be added to the manuscript. Specifications of the CHIKV-VLP used for the screen need to be incorporated into this section and. There are multiple CHIKV clades, this is not addressed during the antibody screening process, the specificity of the antibodies for the different clades needs to be addressed experimentally.
Also, screening against MAYV-VLP represents only one of potential closely related alphaviruses additional experiments to confirm antibody specificity for CHIKV in relation to other viruses is necessary.
3.1 Based on the data in supplemental figure 1 and the screening criteria provided there were many antibodies that could have been used in the assay development, why were the specific antibody pairs chosen. Other data is eluded to, can you include LOD data for all antibodies needs to be shown and flow cytometry screening of infected cells.
Line 201 Citation is needed for the range of E1 and E2 concentration found in acute Chikungunya patients.
3.3 The data presented in the table and use of CT cutoff needs to be better described. As it stands the authors state a CT value of 22 maximizes the performance of the ELISA based on linear regression sensitivity and specificity calculations, but looking at the actual sample numbers, there are substantially more ELISA positive samples than the PCR testing. Their analysis is skewing towards false positives. This is a limitation of the assay that needs to be address or more samples need to be tested to confirm their mathematical modeling. I question the low overall sensitivity of 51%.
Reviewer 3 Report
In this interesting paper by Reddy et al., the authors claim an antigen E1/E2-based ELISA and a lateral flow using a couple of specific monoclonal antibodies for the early detection of Chikungunya virus infection as two alternatives to RT-qPCR. The problem statement is clear and relevant literature has been cited to justify an investigation. The methods used in the manuscript have widespread acceptability in the scientific community and presented with sufficient detail to validate the study. The manuscript is overall well-presented and the data are convincing.
My concerns:
#1. The S1 and S2 figure legends are lacking.
#2. Did the authors validate the two antigen-capture methods using inactivated serum samples by heat treatment or addition of detergent as Triton X-100 ? The safe handling of contaminated human fluid samples such as sera collected from Chikungunya patients in areas with limited laboratory capacities must be taken into consideration.
#3. It would be informative that the authors discuss their diagnosis method based on the capture of free virus particles using specific antibodies in relation with the performances of the recently developed RT-LAMP method that has been recommended for use in infrastructure with poor settings (Loper-Jimena et al., PloS NTD, 2018, e0006448).
Round 2
Reviewer 2 Report
The authors revisions have addressed all of my major concerns.